

# Brief Communication : Mimicking periglacial landforms and processes in an ice-rich layered permafrost system with polydispersed melamine materials : a new concept

Emmanuel Léger[1], François Costard[1], Rémi Lambert[1], Albane Saintenoy[1], Antoine Séjourné[1], and Maxime Leblanc[1]

[1]Université Paris-Saclay, CNRS, GEOPS, 91405, Orsay, France

**Correspondence:** emmanuel.leger@universite-paris-saclay.fr

**Abstract.** This paper presents results on testing polydisperse melamine material versus sand for laboratory ice-rich layered soil system under thawing conditions. We demonstrate the potential of using polydisperse melamine particles in the aim of mimicking the permafrost geomorphological degradations and landslide found in periglacial field environments. We stress that this type of particles, designed for flow and sedimentary processes in river due to their light particle density and the

granulometric size they span, are as well adequate for modeling more realistic geomorphological thawing features observed in cryosphere environments such as slump blocks.

## 1 Introduction

In cryosphere environments the active layer seasonal freeze-thaw cycle on downslope terrains may be associated with a wide range of mass movements, including creep, gelifluction, plug-like flow (Ballantyne, 2018). The mechanisms influenc-

ing these mass-movement phenomena are dependent on soil properties, ice and water contents (Hjort et al., 2014) and thermal regimes (Harris et al., 2001), as well as slope angle (Rowley et al., 2015). Observing and quantifying these phenomena in real environment have been done through remote sensing imagery, e.g. Mithan et al. (2021), while an excellent attempt of direct measurements has been made by Fiolleau et al. (2024) through a dense monitoring approach in Alaska, USA. In parallel, aiming for better constrains on the conditions triggering these destabilization, laboratory-scale analog studies started at the be-

ginning of the 1970's with the study of Higashi and Corte (1971), followed by numerous other experiments at different scales, e.g. Harris et al. (2001), and focusing on various ice-type bodies.

     Among the large typology of cryospheric landforms and processes, one of the most significant markers affected by climate warming are Retrogressive Thaw Slumps (RTS) : landslides caused by the thawing of ice-rich permafrost (Séjourné et al., 2015). As a consequence of the larger amplitude warming in high latitudes, it is projected under RCP4.5 an increase of

RTS formation to above 10,000 per decade after 2075 (Lewkowicz and Way, 2019). These RTS represent large scale active thermokarsts due to the thawing of exposed ground ice in the headwall to form landslide-like U-shaped scars (Lantuit and Pollard, 2008) with wide troughs and slump blocks (Swanson, 2012). The Batagay megaslump in Yakutia (Opel et al., 2019),





with a length of more than 1000 m, a width of 800 m and 60 m high headwalls, represents at the present day the largest and the most active identified RTS in periglacial environments.

The possible causes of present-day initiation of these RTS are not well understood, especially concerning the role of ice-rich permafrost layer on the efficiency of the process and its geometry. In addition the various landforms observed resulting from thawing are difficult to predict. One of the solution goes through laboratory experiments allowing to simulate slope processes within thawing frozen fine-grained soils with the ability to control environmental parameters, precise definition of experimental boundary conditions, and the capability to reduce the time scale of freeze-thaw cycling. This allows many years

of field processes to be modeled in a relatively short time period (Harris et al., 2001). Our work was inspired by the study of Costard et al. (2020), in which the authors pointed out the role of the ice-layer on the destabilization process by reproducing RTS analogs at laboratory scale. We pursue here this work, with this aim of re-creating distinctive geomorphological 3D features observed in large RTS, such as slump blocks, using very light polydispersed plastic particles (PPP). Through this study we aim to address similitude between small-scale laboratory models and field observations. The main objective of the

paper is to evaluate the applicability of this plastic media to recreate specific 3D geomorphological thawing features, notably slump blocks, within laboratory scale.

## 2    Materials and Methods

### 2.1    Thawing-layer material

The laboratory experiments were done with 2 types of material composing a frozen bloc where a pure ice layer was present

in the middle : Fontainebleau Sand considered here as our reference model and the polydispersed plastic particles (herein acronymed PPP). For both material we manipulated the saturated sand/PPP mixtures to a wet bulk density corresponding the "Proctor Compaction" technique (Costard et al., 2020), and froze them. We detail the two different media below.

### 2.1.1    Fontainebleau sand

The reference models were made of frozen saturated Fontainebleau sand from a near-by quarry, sold to be made of 99.9%

silicium. The sand's physical and hydrodynamic parameters, such as the Mualem (1976) - van Genuchten (1980)'s parameters ruling the hydrostatic and hydrodynamic can be found in Léger et al. (2020) and two of them, $\alpha$ ruling the capillary height and inversely proportional to the air entry value and $K_s$ the saturated hydraulic conductivity are summarized in the Table 1. The particle size distribution (PSD) was determined using a Mastersizer 3000 HYDRO-G (Malvern Panalytical Ltd.) laser diffraction particle size analyzer delivering liquid dispersion with an extended range of the particles spanning from 0.01 to

3500 $\mu$m. The PSD is presented in Figure 1-a) with cross symbology. As a comparison we show as well the PSD for a Yakutian sand extracted from Gautier and Costard (2000) showing the close similarity with our laboratory sand. The Fontainebleau sand is particularly well defined in term of grain size, having a $C_u = \frac{d_{60}}{d_{10}}$ ratio at 1.76, being poorly sorted while the Yakutian sand is about 1.3-1.4. The bulk density and porosity for the Fontainebleau sand were measured on samples using the same compaction





**Figure 1.** Experimental setup : a) granulometric curves for the polydispersed granular media PPP, the Fontainebleau sand (France) and the sand from Central Yakutia (Gautier and Costard, 2000). The different colors represent different samples while the dashed and plain lines are sigmoidal fitting. b) Schematic of the experiment, side view, where the granular media is set on top of the cooling coil at 17° angle relative to horizontal, the thermistors in yellow circle at the center are labeled C while on the side labeled S; c) upward view of the full experiment with nadir and side cameras (1), granular material (30 cm wide x 40 cm long x 15 cm height) and thermistor strings (2) and (3) freezing coil covered with ice; d) zoom-in of the cuboid of polydispersed color-coded-by-size material, before starting the thawing phase. The cable are linking the thermistors rods (S and C) to the dataloger.

used for the experiment giving $1.7 \pm 0.1$ g/cm$^3$ and $0.39 \pm 0.02$ for porosity. The hydrodynamic and physical parameters are

summarized in Table 1 under the label "Font. Sand".





**Table 1.** Uniformity coefficient ($C_u$), particle and bulk densities ($\rho_s$ and $\rho_b$) and Mualem (1976) - van Genuchten (1980)'s hydrodynamic parameters for the Fontainebleau sand and the polydispersed plastic media ($\alpha$ being proportional of the inverse of air-entry value and saturated hydraulic conductivity, $K_s$).* are from Léger et al. (2020)

| Parameters | PPP | Font. Sand |
|:---:|:---:|:---:|
| $C_u = \frac{d_{60}}{d_{10}}$ | 2.0 | 1.76 |
| $\rho_s$ [g.cm$^{-3}$] | 1.7±0.1 | 2.7±0.1* |
| $\rho_b$ [g.cm$^{-3}$] | 0.77± 0.02 | 1.7± 0.1 |
| $\phi$ | 0.55± 0.1 | 0.37 ± 0.1 |
| $\alpha$ [cm$^{-1}$] | 0.069±0.006 | 0.024± 0.002* |
| $K_s$ [cm.min$^{-1}$] | 7.7±0.1 | 1.25±0.25* |

### 2.1.2 Polydispersed plastic particles (PPP)

We based our study on the Hughes (1993)'s suggestion to use non-cohesive sediment with different densities at a lower scale for mimicking fluvial dynamic of erosional and sedimentary processes observed in river at laboratory macro-scale. Here the novelty resides in using these particles to re-create cryo-induced geomorphological features. These polydispersed melamine particles (made from recycled plastic (Emriver, Carbondale, IL, USA)) were tested as frozen layer material. Specific color were associated with different particle size according to the manufacturer : red 0.4 mm, brown 0.7 mm, white 1 mm and yellow 1.4 mm for visualization purpose. These PPP have more realistic mobility than fine sand particles at the scale of our simulation, that makes possible to observe geomorphological analogs (Hughes, 1993) within a reasonable amount of experimental time. Since the polydispersity could induce an heterogeneous mixing, we performed particle size analysis using sieves and the resulting PSD curves are presented in Figure 1-a) with filled circles. The uniformity coefficient, $C_u = \frac{d_{60}}{d_{10}}$, were calculated on five different samples, giving an average value of 2.04, being synonymous of not well graded soil. The bulk density and porosity for the PPP were measured on samples with the same compaction than the one used for the box experiment giving 0.77 ± 0.02 g/cm$^3$ and 0.50 ± 0.02 for porosity. The saturated hydraulic conductivity, $K_s$, was determined using a "Ksat" apparatus (Hyprop, Metergroup, Pulmann, WA) giving large values of saturated hydraulic conductivity, $K_s = 1.29*10^{-3}$ m/s, almost one order of magnitude higher than the Fontainebleau sand one. The Mualem (1976) - van Genuchten (1980)'s parameter fitting the soil-water retention function were determined using the Rosetta algorithm (Schaap et al., 2001) from the PSD. For PPP media, the $\alpha$ parameter was equal to 0.069±0.006 cm$^{-1}$, i.e. almost 3 threefold the value for Fontainebleau sand, synonymous of a factor 3 in the capillary height/air-entry value. The parameters are summarized in Table 1 under the label "PPP".

### 2.2 Experimental setup

We used the cold room at the GEOPS laboratory (Paris-Saclay University, France) dedicated to the physical modeling in periglacial geomorphology (Costard et al., 2020; Léger et al., 2023). The specificity of the presented experiments resides on simulating the thawing of ice-rich frozen soil with an pure ice-layer inside set on top of permafrost simulated with a cryostat



coil at -10 °C to insure the thermal homogeneity of the frozen sample during the first few hours of the initialization phase. The frozen layer was built with fully water saturated media ( representing 0.55 % and 0.37 % of the volume for PPP and

Fontainebleau sand respectively), priory frozen in a freezer at -18 °C in a cuboid plastic mold during 72 hours, then set on top of the -10 °C permafrost table tilted at 17° (see Figure 1-b, c and d). The obtained frozen material was a parallelepiped block of 30 cm wide x 40 cm long and 15 cm thick (see Fig 1-b,c), it was set on a the dipping layer (17°) above the cryostat at -10°C in a cold chamber maintained at atmospheric temperature (15-20°C). The cryostat was maintained on for few hours during the initialization phase and then shut off during the thawing process, for simulating a deepening of the active layer, and

accelerating the thawing. Two types of experiment were performed with both materials, Fontainebleau sand and PPP: layers thawing composed from (i) frozen saturated Sand/PPP sandwiching a 5 cm of pure ice layer (5 cm Sand/PPP - 5 cm ice - 5 cm Sand/PPP) presented here, and ii) frozen saturated homogeneous Sand/PPP (15-cm thick) available in the supplementary materials.

## 2.3 Temperature and optical monitoring

Two thermal monitoring positions were set every 2 vertical cm (with the origin at the permafrost table, the position are : 2 cm, 4 cm, 6 cm, 8 cm and 10 cm height) in the center (labeled C) and in the up-side (labeled S) of the block to quantify the side effects (see Figure 1-b, c and d)). The sensors, a string of 10 thermistors (PT100 with $\pm$ 0.1 °C), monitored the block temperature at 10 minute intervals. The sensors were set inside the block before the initial freezing.

During the experiments, the block surface deformation was optically monitored with 2 Canon EOS 550 D cameras triggered

with Raspberry-Pi based time-lapse trigger (Witty-Pi board (UUGear) + gphoto2 library) at 30 min interval (See the placement in Figure 1-c). These measurements were carried until geomorphological activity ceased.

## 2.4 Degradation quantification

The degradation was measured based on the block surface variations observed at almost nadir angle using one of the camera (camera labeled (1) in Figure 1-c). The degradation was compiled relative to the initial surface area in term of area change

percentages. We measured the area of each detached slump blocks through time as well as the initial block deformation (Figures 2-A/ and B/). We are aware that we were creating a bias on the degradation between the two thermal measurement rods (See Fig 1-b, c and d), but no technical solution was found to maintain the sensor positions without keeping a vertical rod in place. We stress as well, that we are aware of the thermal side effects on the block, however as all the experiments were carried out with the same bias, the inter-comparison was possible. We used QGis (QGIS Geographic Information System,

www.qgis.org) to geo-reference, with local geometrical projected system, each picture by applying the same ortho-rectification on them based on the first picture (known initial dimension of 30x40x15 cm$^3$) and the measuring tapes present in pictures. We chose 8 snapshots on each experiment to calculate the non-degraded area left in place as well as the degraded block area. As the pixel size represents 0.2 mm x 0.2 mm, we over-estimated the uncertainty by assuming 20-pixel uncertainty on each pick propagating the uncertainty on the surface computation.



## 3 Results

A total of 10 experiments have been carried out, testing the material-water mixing ratio, the time and space scales and the time needed to freeze and thaw. In this paper we present in detail only one set of experimental results for PPP and Fontanebleau sand materials and one scenario (ice-layer cases), while the homogeneous cases are presented in the suppl. materials to illustrate the role of the ice-layer on destabilization. The degradation experiments are presented in Figure 2-A and Figure 2-B for Fontainebleau sand and PPP material respectively. On each sub-figure in Figure 2 A and B, the inset i) represents the temperature time series measured in the block (side and center), while insets ii)-ix) are snapshots monitoring the degradation at 8 timeslices.

### 3.1 Fontainebleau sand experiment

#### 3.1.1 Temperature and degradation monitoring

The temperature time series measured for the Fontainebleau sand reference block are presented in Figure 2-A-i), accompanied with the colored degraded area at 8 different time lapses and two pictures from which are calculated the degraded areas as example (presented in Figure 2-A-ii) to ix)). On the thermal time series, Figures 2-A-i), three main stages were recorded during the warming phase. The block started at -18°C placed over the cryostat at -10°C followed by a rapid increase of the block temperature up to the phase change (approximately the first 5 hours). The second stage corresponds to the latent heat release with a typical zero-curtain effect (between 5 and 22 hours). The third stage starts at the stop of the basal cooling coil ("cryo-off" at 22 hours presented in Figure 2-A) with a subsequent warming of the block up to positive temperatures. As expected the center sensors (C2 to C10) are following a less steep temperature increase rate than the side sensors (S2 to S10). The experiment was shut down when geomorphological activity ceased completely . The forms of degradation exhibited (Figure 2-A-ii) to ix) and especially the pictures in xi-b) are similar to RTS phenomena, where the vertical structure of the detached block is not kept and horizontal translation is strong. In addition a vertical subsidence is observed on the main block, due to the water draining out of it, visible in snapshot at 22.4 h in Figure 2-A-ix-b). The homogeneous case, presented in the Supp. materials did not present any geomorphological activity nor deterioration during the experiment.

### 3.2 PPP experiment

#### 3.2.1 Temperature and degradation monitoring

The temperature time series measured for the PPP block are presented in Figures 2-B, accompanied with the colored degraded area at 8 different time lapses and two pictures from which are calculated the degraded areas (Figures 2-B ii) to ix)). Similar to what have been observed for the reference Fontainebleau sand case, three main stages during the thawing phase were recorded. In the first stage, we observe a rapid increase of the block temperature up to the phase change (approximately the first 8 hours). The second stage corresponds to the latent heat release with a typical zero-curtain effect (between 8 and 55 hours for the ice-layer case presented in Figure 2-B). The third stage starts at the stop of the basal cooling coil ("cryo-off" at 55 hours



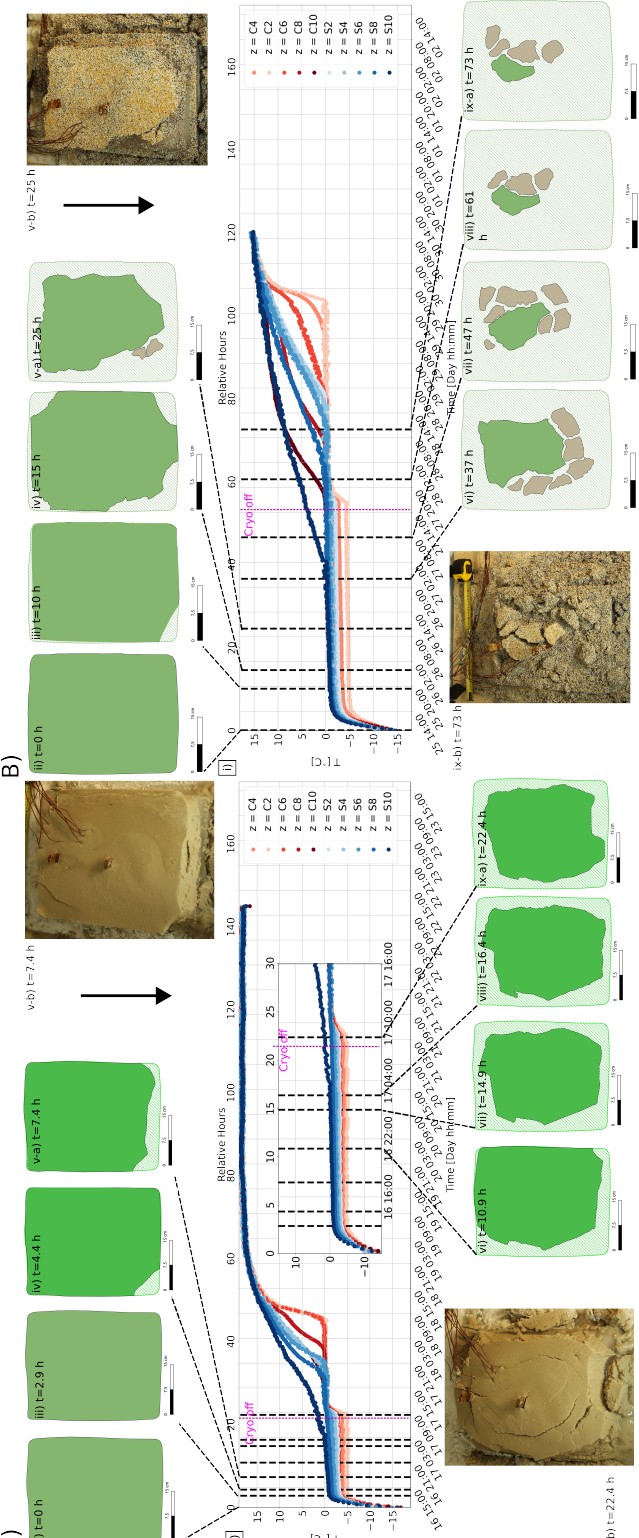

**Figure 2.** A/ i)Thermal and ii-ix)degradation time series for the ice-layer Fontainebleau sand case. In i) red colormap : center thermal monitoring positions in centimeters, in blue colormap: upper side monitoring position in centimeters. The "cryo-off" purple vertical line symbolize the stop of the cold basal coil. Drafts ii-ix) symbolize degradation area extracted from snapshots, where the stripped green is the initial area, the green is what is left in place at each time-lapse. Two pictures (v-b and ix-b) from which are extracted the degradation area at 7.4 h and 22.4 h are shown. The black arrow symbolizes slope direction. B/ i)Thermal and ii-ix)degradation time series for the ice-layer case using PPP. In red colormap : center thermal monitoring positions in centimeters, in blue colormap: upper side monitoring position in centimeters. The "cryo-off" purple vertical line symbolize the stop of the cold basal coil. Drafts symbolize degradation area extracted from snapshots, where the stripped green is the initial area, the green is what is left in place at each time-lapse and the light gray correspond to the detached slump block. Two pictures from which are extracted the degradation area at 25 h and 73 h are shown. A side close-up is available in supplementary information Figure S1





presented in Figure 2-B) with a subsequent warming of the block up to positive temperatures. In all cases, the experiment was shut down when geomorphological activity ceased. Similar to what has been observed for the Fontainebleau sand cases, the temperature is warmer on the side (in blue) of the block compared to the center, as expected (warming faster on the sides) due to side effects. The geomorphological degradation is strong and illustrated by vertical block separated from the main block.

The vertical structure of the block is kept for a certain time before it looses its cohesion. These block are similar to slump blocks observed in the field (Swanson, 2012). They start to appear after 10 h of experiment, even before the shutdown of the cooling coil. Comparing with the homogeneous case in Supp. Material, the temperature dynamic seems to be similar, in term of thawing. One can observe, however, that the thawing and the destabilization is faster in the ice-layer case indicating that the ice layer is disturbing the integrity of the block by decreasing its cohesion (Costard et al., 2020).

**3.3    Fontainebleau sand versus PPP experiment comparisons**

For both materials, there is stable phase after the "cryo-off" and then rapid increase of soil temperature, notably faster for frozen sand (with steeper thermal curves) than the ones for frozen PPP (see Figures 2-A and 2-B). This stable phase after the cryo-off can be explained by the presence of the ice layer which maintains a low temperature as already pointed out by Costard et al. (2020) and by comparing with the homogeneous cases presented in supplementary materials. We calculated the onset and

the evolution of degradation for each experimental setup using the area percentage extracted from snapshots. The degradation evolution for the two types of shown experiments (the two homogeneous cases are available in Supp. Materials) is summarized in Figure 3. The degradation is dramatic for the PPP comparing with the Fontainebleau sand case. The ice-layer induces a longer onset of degradation than the one for the homogeneous model for sand and PPP. In more details, it appears that the impact of the warming phase is more progressive with the PPP than for the sand experiment. Beyond the degradation the

geomorphological features observed wit the PPP is similar to slump blocks, while the sand thawing do not allow to create such features (A side view zoomed in the slump block is visible in supplementary materials Figure S1).

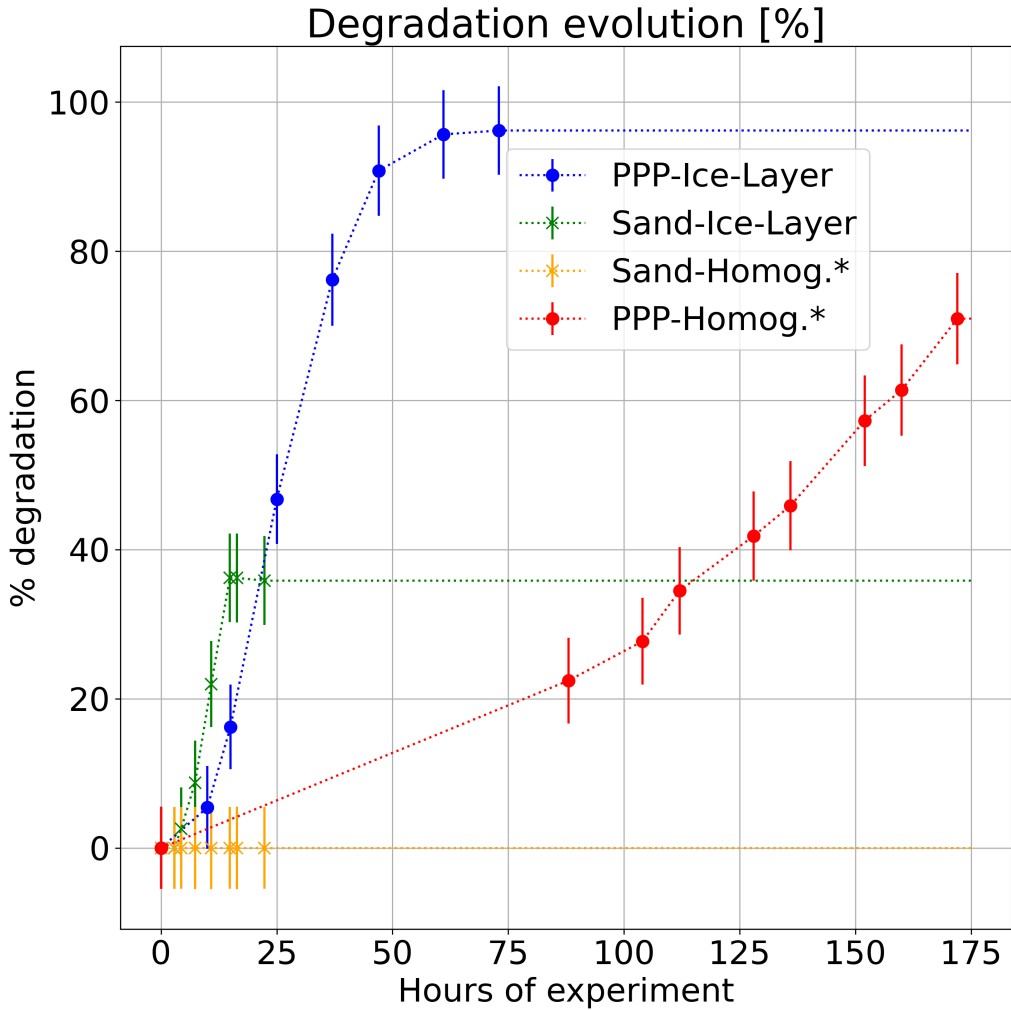

**Figure 3.** Percentage of degradation through time for the four types of experiment. Filled circles correspond to the PPP experiments (red for the homogeneous case in Supp. Material and blue for the ice-layer cases in Figure 2-B) while the crosses result from the sand experiments (yellow for the homogeneous case in Supp. Material and green for the ice-layer case in Figure 2-A). The * mark experiments available in suppl. materials.

## 4   Discussions

The experiments using PPP induce a larger thermal degradation than those done with the reference sand model, and most importantly allows to simulate the creation of a fracture network and slump blocks, features not present in the sand experiment
(See Figure 2-B and Figure S1). The presence of an ice-layer induces a higher onset of erosional degradation (earlier start and





longer phase of degradation) than the one for the homogeneous model due to the melting of the ice layer, as already pointed out by Costard et al. (2020).

The size, geometry and ice-layer within the frozen soil represent a simplification of the natural settings, but the main objective here was to test the influence of lightweight material for on permafrost degradation from a 3D geomorphological points of view. The reference model (Fontainebleau sand) exhibits a slight evolution mostly based on destabilisation processes, while the heterogeneous one with PPP showed dramatic evolutions (slump blocks).

The analysis of the time-lapse images showed that the evolution of the individual slump block towards a rather hummocky morphology similar to the development of RTS in natural periglacial environments. Our experiments clearly show the individual detachments and vertical subsidence of individual blocks and the subsequent formation of eboulisation slope due to the cohesionless particles during thawing of the ice-rich frozen sample.

In these experiments, we assumed that the scale effect was not a limiting factor. The size of the block allows reasonable side effect. During the warming phase, the collapsed blocks along the sides remain low (up to 2 cm from the side). Here, we restricted our approach to the relative importance of the PPP material versus fine sand material and their ability to produce 3D degradation morphology like the ones observed along the RTS on the field.

Beyond the PSD difference between the two types of used material, we clearly benefit from the size variability for the PPP versus the very narrow defined PSD of the Fontainebleau sand, we stress that the saturated hydraulic conductivity, $K_s$, and capillary forces ($\alpha$ and air-entry values) play a major role in the degradation phenomena we observe. Indeed there is almost one order of magnitude in the saturated hydraulic function and 300% difference in the $\alpha$ parameter synonymous of air-entry value. These large difference in the water retention, favor the creation of fractures for the PPP media, creating more easily the slump blocks. In addition the bulk density difference allows to create phenomena only observed in the case of clay particle in active layer. We point out as well the potential role of electrostatic forces between plastic particles, mimicking at a certain extend what we observe in the case of clay layer and probably allows the cohesion of the block through time. We do believe that using this type of plastic/melamine particles is an excellent alternative for modeling geomorphologically cryo-induced degradation. Our laboratory results are in agreement with recent investigations along the Batagay thaw slump in Yakutia, where the presence of icy layer is the precondition for its deep incision (Opel et al., 2019).

## 5 Conclusion

Our experiments demonstrate the ability of poly-dispersed plastic material to replicate specific features of permafrost degradation, notably slump blocks, at laboratory scale within a reasonable amount of experimental time. We compared it with classical sand used in laboratory analogs and different interfaces (homogeneous, ice-layer) to observe the geomorphological advantages to use such particles. By their light density and peculiar hydrodynamic parameters we showed that new geomorphological cryo-induced features could be modeled at laboratory scale within thawing experiment.

We demonstrate the potential of using polydisperse melamine particles in the aim of mimicking the permafrost degradation and the landslide processes found in periglacial environments. The use of PPP media demonstrate its ability to correctly mimic



typical periglacial morphologies in 3D such as slump blocks under warming condition with a relatively short time-scale. Even if the permafrost thawing process is simplified in our experiments, we have highlighted how the presence of an ice layer deeply influence the efficiency of the thermokarst phase that are qualitatively relevant to the evolution of periglacial systems under recent warming. Further investigations are needed to determine the scaling relations and the link with a 3D numerical modeling.



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

*Author contributions.*  E.L., F.C. and R.L. performed the cold room experiments, E.L., R.L., A.Sa and M.L. performed the lab experiments; E.L., F.C. and R.L. performed Data curation; E.L., F.C., R.L. and A.Se did the formal analysis, E.L. and F.C. brought experimental resources, E.L. and F.C. performed the funding acquisition. E.L. and F.C. wrote the original draft and all the authors conducted the reviews and editing.

*Competing interests.*  The contact author has declared that none of the authors has any competing interests.

*Acknowledgements.*  The author thank the PANOPLY analytical platform. This article is dedicated to Élie Lambert-Bieuzent.