# Peer review of "Brief Communication: Mimicking periglacial landforms and processes in an ice-rich layered permafrost system with polydispersed melamine materials: a new concept"

_EGUsphere, 2024_

## Author Response (AR1)

We express our gratitude to the reviewer for their constructive suggestions, all of which have significantly enhanced the quality of our manuscript. Below, we provide our responses to the main comments. The original reviewer's comments are presented in italics, while our responses are in standard font.

**1 Reviewer 1**

- *1. The abstract is easy to follow overall. What does 'mimicking' mean, experimental evaluation or numerical simulation?*
  We rephrased the abstract with the adjective "experimentally" mimicking to avoid confusion. (Line 3).

- *2. The limitations of laboratory experiments are not well described. Therefore, the motivation for using PPP is not clear; please revise it.*
  We added the following sentence (Lines 30-37) :
  " Most of the laboratory experimental studies, when done with one component mixture for the thawing material soil, are done with sand. In the case of lightweight particles, clay or silt can be considered, but lead to experimental difficulties due to their volatility and compactness after the melting, since going back to their fine powder initial version requires long drying and crushing. We consider here the solution brought by PPP particles, due to their realistic mobility than fine sand particles at the scale of our simulation. We point out as well the potential role of electrostatic forces between plastic particles, mimicking what we observe in the case of clay and silt materials. In addition, these types of particles are granulometric-size color coded and as such help the visualization of cryo-induced morphological features."
  *3. What are the benefits of using PPP compared to conventional sand/soil materials?*
  Answered above and this is one of the conclusions of the paper as well.

- *4. Figure 1 is excellent.*
  Thank you

- *5. The size distribution of PPP will significantly influence the flow and sedimentary behaviour. Please explain the specific particle size selection criteria.*
  This is good remark, thank you. We added a sentence in this direction (Lines 185-187):
  "The study of the size influence on the flow and sedimentary behavior is beyond the scope of this study. We focused, here, on mimicking experimentally special geomorphological features using PPP as a first test."

- *6. The authors mentioned potential future perspectives, such as exploring scaling relations and 3D numerical modelling. Please be more specific.*
  We added a sentence in conclusion to be more specific (Lines 210-214):
  "Further investigations are needed to determine the scaling relations between parameters, which allow the transpose of similarity between model and prototype. This implies the use of dimensionless parameters, such as the densimetric Froude number, the grain size Reynolds number and the Engelund–Hansen formula, similar between the analog and the field model. We plan as well to complement the

surface change with time-lapse 3D reconstruction, using photogrammetry or laser scanning."(Lines 210-214)

**2 Reviewer 2**

Dear reviewer,

Thank you for your very constructive and high scientific level comment you wrote.

The paper was thought of as a preliminary approach on using PPP to mimic some typical periglacial morphologies observed in arctic regions. We wanted to use the peculiar PPP hydrodynamical properties, and their light weight to trigger geomorphological features we were not able to observe with sand. This is why we aimed for a brief communication issue as a first study. We are aware that the size is not the main factor affecting the analog of the RTS at lab scale, and this is why we concluded the paper with the following sentence (Following R1 remarks): (Lines 210-214) "Further investigations are needed to determine the scaling relations between parameters, which allow the transpose of similarity between model and prototype. This implies the use of dimensionless parameters, such as the densimetric Froude number, the grain size Reynolds number and the Engelund–Hansen formula, similar between the analog and the field model. We plan as well to complement the surface change with time-lapse 3D reconstruction, using photogrammetry or laser scanning."

- "*Such differences are not simply size, but may change the relevant phenomena that control RTS. Quite old research (Lewkowitz) suggests that positive pore-pressures developed during thaw play a strong role in the progression of RTs, for example. Therefore, the weight of the soil at depth is important. Relevant dynamics of thaw vs flow and their differences across scale should also be considered*". As you mentioned, the scale of the landslide increased as the depth was increasing in the laboratory simulation, but some other parameters could be important such as the depth of the active layer, the ice content. We did not measure the pore pressure for these experiments as we were doing very preliminary research. This will be the object of future studies.

- "*The authors comment on the differences in hydraulic properties and seem to imply these are dominant. My first reaction upon reading the document is that the mechanical property difference needs to quantified, as strength loss / volume change in in the much looser PPP samples would be much greater than for the sand. As someone who would like to try simulation this kind of test, that information would be required*" Concerning the hydraulic properties, these were the easiest to measure for both types of material and were an indication of the hydrodynamical contrast between these two materials. The mechanical properties necessitate more time, as well as any rheological analysis. Such a topic needs further developments in our laboratory and much more experiments are necessary in the near future.

- "*The authors comment on the likely mobility of the PPP material. I presume they mean the rheology, which is important as a risk posed by RTS is the run-out, sometimes to considerable distances, of the thawed soil. More information on the rheological properties of the thawed sand and PPP, and how these compare with rheology inferred from field observations, would be of interest*". We

meant the mobility of the particles, because we were able to track them in the images. This is, in a way, a consequence of rheology. Once again, the purpose of this brief communication was to visually observe peculiar geomorphological phenomena that are not classically observed in sand experiments Based on your comments, we have added the following sentence to the conclusion : Lines 214-215 "We are aware that the role of positive pore pressure is crucial for the development of RTS and that the rheological properties of sand vs. PPP are needed to fully quantify the dynamics of thaw vs. flow. This will be the subject of further studies."

Kind regards
Emmanuel Léger et co-authors